# Lmser-pix2seq: learning stable sketch representations for Sketch healing

## Abstract

Sketch healing aims to recreate a complete sketch from the corrupted one. The sparse and abstract nature of the sketch makes it challenging. The features extracted from the corrupted sketch may be inconsistent with the ones from the corresponding full sketch. In this paper, we present Lmser-pix2seq to learn stable sketch representations against the missing information by employing a Least mean square error reconstruction (Lmser) block, which falls into encoder-decoder paradigm. Taking as input a corrupted sketch, the Lmser encoder computes the embeddings of structural patterns of the input, while the decoder reconstructs the complete sketch from the embeddings. We build bi-directional skip connections between the encoder and the decoder in our Lmser block. The feedback connections enable recurrent paths to receive more information about the reconstructed sketch produced by the decoder, which helps the encoder extract stable sketch features. The features captured by the Lmser block are eventually fed into a recurrent neural network decoder to recreate the sketches. Experimental results show that our Lmser-pix2seq outperforms the state-of-the-art methods in sketch healing, especially when the sketches are heavily masked or corrupted.

## 1 Introduction

Humans are able to complete things that are missing in life through their imagination, such as completing blanks, novel sequels and image repairs. Sketch healing task (Su et al., 2020) is one of the related works. Sketch healing is to synthesise a complete sketch that best resembles the partial input (Su et al., 2020; Qi et al., 2022). Different from the image inpainting task (Pathak et al., 2016), where photos have rich texture information, freehand sketches are highly abstract and sparse, making sketch healing quite challenging.

The way to get a corrupted sketch, proposed by Su et al. (2020), is to crop several local visual patches from a raster sketch image and drop some of them. This approach results in a corrupted sketch raster image and some remaining visual patches. Conventional sketch generation models (Chen et al., 2017; Zang et al., 2021) that take images as input can be used for sketch healing. However, these models designed for sketch synthesis are not comparable to SketchHealer-1.0 (Su et al., 2020), which was specifically designed for sketch healing. SketchHealer-1.0 constructs a graphical representation of the sketch by treating patches as nodes and connecting edges based on the nodes' temporal proximity, i.e., the drawing order. The graphic sketch representation realizes the information interaction between different patches in the same sketch, so as to achieve a better effect of healing. Based on SketchHealer-1.0, SketchHealer-2.0 (Qi et al., 2022) considered the relationship between the local reconstruction and the global semantic preservation. SketchHealer-2.0 requires the involvement of a pre-trained model to calculate the semantic similarity between the recreated sketch and the full sketch. SketchHealer-1.0 (Su et al., 2020) and SketchHealer-2.0 (Qi et al., 2022) build graphs that depend on drawing order, but this information is not always available. To overcome this difficulty, SketchLattice (Qi et al., 2021) proposes a novel lattice representation and takes image as input. However, during the data processing phase, the lattice approach causes some of the information in the raster sketch image to be lost, thus limiting SketchLattice's performance.

Different from the state-of-the-art graph-structure models, which pass information between nodes to fill in the gaps, we expect that the network to take full advantage of the information in the raster sketch images and learn stable sketch representations in the absence of temporal information. Stable representations mean that the model extracts the features of the corrupted sketch as consistent

as possible to the ones of the full sketch. Theoretically, this consistency allows different corrupted sketches obtained by masking from a full sketch to be recreated similarly. Conversely, when the extracted features are unstable (lack of consistency), the healed sketch fails to maintain semantics, and worse, its category changes.

To learn stable sketch representations, we expect the feature maps from different layers in the network to be fully fused, which helps to extract significant and stable features from the corrupted sketches. Least mean square error reconstruction (Lmser) (Xu, 1991; 1993) enables this purpose. Lmser was a development of autoencoder (AE) (Bourlard & Kamp, 1988) by folding and merging the symmetrical encoder and decoder together. Such folding is equivalent to adding bi-directional skip connections between the encoder and the decoder (Xu, 2019). The effectiveness of Lmser is demonstrated in image inpainting (Huang et al., 2020b), super-resolution (Li et al., 2019), and semantic segmentation (Guo et al., 2019; Cao et al., 2021). However, these studies focus on image-related applications with rich texture information, rather than sparse and abstract sketches.

We present Lmser-pix2seq to learn stable sketch representations against the missing information by employing a Lmser block, which falls into encoder-decoder paradigm. Taking as input a corrupted sketch, the Lmser encoder computes the embeddings of structural patterns of the input, while the decoder reconstructs the complete sketch from the embeddings. We build bi-directional skip connections between the encoder and the decoder in our Lmser block. The feedback connections enable recurrent paths to receive more information about the reconstructed sketch produced by the decoder, which helps the encoder extract stable sketch features. The features captured by the Lmser block are eventually fed into a Recurrent Neural Network (RNN) decoder to recreate the sketches.

In summary, our contribution is that we propose Lmser-pix2seq to learn stable sketch representations for sketch healing. The bi-directional skip connections in our Lmser blocks allow the feature maps from the encoder and decoder to be sufficiently fused to facilitate the extraction of sketch features. Experimental results show that our Lmser-pix2seq outperforms the state-of-the-art methods, especially when the sketches are heavily masked or corrupted.

## 2 RELATED WORK

**Sketch Generation.** Research related to sketch generation with deep learning methods (Ha & Eck, 2018; Zhou et al., 2018a; Das et al., 2021; Ge et al., 2021) has been developing rapidly in recent years. An interesting work on sketch generation is that the neural network imitates humans to draw the vector sketch stroke by stroke. sketch-rnn (Ha & Eck, 2018) is an RNN-RNN architecture generation model based on the Variational Autoencoder (VAE) (Kingma & Welling, 2013), which enables the conditional and the unconditional single category sketch generation. Later, the proposed sketch-pix2seq (Chen et al., 2017) with the convolutional neural network (CNN) encoder solves the multi-category generation problem and finds that latent code with the normal distribution constraint removed has better reconstruction results. Inspired by the above two models, (Song et al., 2018) fuses photo texture information with temporal information by shortcut cycle consistency. To further improve the controllability of the generation, RPCL-pix2seq (Zang et al., 2021) assumes that the latent space follows a Gaussian mixture model (GMM), and the number of Gaussians is determined by automatic selection of the model. sketch-rnn and RPCL-pix2seq are more similar to our model. There are some large pre-trained models, e.g. Sketch-Bert (Lin et al., 2020) and Sketch-former (Ribeiro et al., 2020), for not only the sketch generation, but also for other downstream tasks. Sketch healing is similar to the combination of vector sketch generation and image inpainting.

**Sketch Healing.** SketchHealer-1.0 (Su et al., 2020) clarified the definition of the sketch healing and proposed a novel graph representation method. SketchHealer-2.0 (Qi et al., 2022) rasterizes the generated sequence and calculates its semantic perceptual loss from the corresponding full sketch. The other model that represents sketch as a graph is the SketchLattice (Qi et al., 2021). SketchLattice is a lightweight network that can construct graphs without relying on the drawing order. SketchLattice lattices the image, then treats the intersection of the lattice with the pixels of the sketch stroke as the nodes, and constructs the edges between the nodes by Euclidean distance. In contrast to the sketch generation task, the sketch healing task requires the model to extract accurate and effective features when the sketch is masked.

**Skip Connection.** Forward connections refer to the direct transmission of information from the shallow layers to the deep layers via a short-circuit path. Studies have shown that forward connections can alleviate the gradient vanishing problem (He et al., 2016) and promote multi-scale feature fusion (Ronneberger et al., 2015). This technique are widely applied in the field of computer vision,

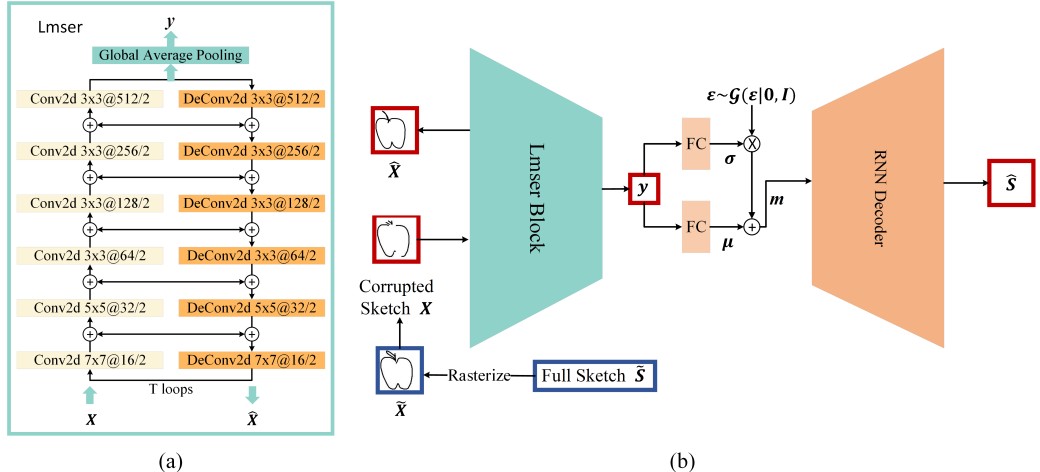

Figure 1: (a) Architecture of Lmser block. The convolution and deconvolution configurations are shown as $h \times w@d/s$, where h, w, d and s represent height, width, depth and stride, respectively. All the layers are followed by an instance norm layer and a relu activation function, except for the bottom-most deconvolution layer, which employs a tanh activation function. (b) overview of Lmser-pix2seq. The latent vector $y$ output at the top of the Lmser block is eventually fed to an RNN decoder to recreate the vector sketch $\hat{S}$.

e.g., image recognition (He et al., 2016; Huang et al., 2017; Dosovitskiy et al., 2020) and image semantic segmentation (Ronneberger et al., 2015; Milletari et al., 2016; Zhou et al., 2018b; Huang et al., 2020a). Feedback connections, as opposed to forward connections, have received attention recently. The implementation of a feedback connection usually requires the assistance of a recurrent mechanism (Xiang et al., 2020; Huang et al., 2020b). Studies on semantic segmentation of medical images have shown that feedback connections can acquire finer details to effectively improve the accuracy of segmentation (Xiang et al., 2020; Cao et al., 2021). In addition, block reuse reduces the number of parameters in the network.

## 3 METHODOLOGY

### 3.1 OVERVIEW OF LMSER-PIX2SEQ

The proposed Lmser-pix2seq is a model consisting of an Lmser block encoder and an RNN decoder, where the Lmser block is also in encoder-decoder architecture. The bi-directional skip connections in our Lmser block allow the feature maps to be fully fused to extract stable features from the sketch. Fig. 1(b) shows the pipeline of our Lmser-pix2seq. A corrupted sketch $X$ is fed into the Lmser block to obtain the reconstructed raster image $\hat{X}$ and the latent vector $y$, respectively. We project $y$ into two vectors $\mu$ and $\sigma$ and then get the final latent code $m$ by the reparametrisation trick (Kingma & Welling, 2013), which is fed into a Long Short-Term Memory (LSTM) (Hochreiter & Schmidhuber, 1997) decoder to output the recreated sequence $\hat{S}$. Specifically, the LSTM predicts the pen state and produces parameters to form a GMM to estimate the offset distance from previous point. Our LSTM decoder with 512 nodes is consistent with the SketchHealer-1.0 (Su et al., 2020). The LSTM modeling method is referenced from the sketch-rnn (Ha & Eck, 2018), from which readers can access more details.

In our method, a sketch has sequence $\tilde{S}(\tilde{s_1}, \tilde{s_2}, ..., \tilde{s_n})$ and raster image $\tilde{X}$ two different modal representations. A sequential representation of the sketch is composed of a series of points, each of which is a vector containing five elements: $(\Delta x, \Delta y, p_1, p_2, p_3)$. $\Delta x$ and $\Delta y$ are the offset distance from the previous point in the $x$ and $y$ direction. $(p_1, p_2, p_3)$ indicates the current stroke state, where $(1, 0, 0)$, $(0, 1, 0)$, $(0, 0, 1)$ represent the three stroke states of touch, lift, and the end of sketch, respectively. $\tilde{X}$ is rasterized by the sequence $\tilde{S}$. The corrupted sketch $X$ is obtained by masking part of the information in the full sketch $\tilde{X}$.

## 3.2 LMSER BLOCK ARCHITECTURE

As mentioned before, we expect a network to recover incomplete sketch raster images while extracting features. We take a convolutional autoencoder (CAE) as the backbone network to implement our Lmser block. The encoder and decoder of the CAE consist of several convolution and deconvolution layers, respectively, as shown in Fig. 1(a). The encoder and decoder layers of the CAE are fully symmetrical. We add bi-directional skip connections to the corresponding layers between the encoder and the decoder. We connect the decoder and the encoder to form a loop, as illustrated in Fig. 1(a), that allows the feedback skip connections to work (Huang et al., 2020b). With the recurrent mechanism, we calculate the feature fusion as follows:

$$
\begin{aligned}
\boldsymbol{z}_l^{te} &= f^l(\boldsymbol{z}_{l-1}^{te}), & t &= 0, 0 < l < L, \\
\boldsymbol{z}_l^{te} &= \alpha f^l(\boldsymbol{z}_{l-1}^{te}) + (1-\alpha)\boldsymbol{z}_l^{(t-1)d}, & t &> 0, 0 < l < L, \\
\boldsymbol{z}_l^{te} &= f^l(\boldsymbol{z}_{l-1}^{te}), & t &\geq 0, l = L, \\
\boldsymbol{z}_l^{td} &= g^{(l+1)}(\boldsymbol{z}_{l+1}^{td}), & t &\geq 0, l = 0, \\
\boldsymbol{z}_l^{td} &= \alpha g^{(l+1)}(\boldsymbol{z}_{l+1}^{td}) + (1-\alpha)\boldsymbol{z}_l^{te}, & t &\geq 0, 0 < l < L,
\end{aligned}
\tag{1}
$$

where $\boldsymbol{z}_l^{te}$ and $\boldsymbol{z}_l^{td}$ respectively denote the $t$-th update of the $l$-th layer's feature maps from the encoder and the decoder, $f^l(\cdot)$ and $g^l(\cdot)$ separately denote the $l$-th layer network-related calculations for the of encoder and the decoder. $\alpha$ is a hyper-parameter, which we set to $0.5$ in the experiments. We use the bottom-most output $\boldsymbol{z}_0^{td}$ of the decoder as the input to the encoder to form the loop, i.e., $\boldsymbol{z}_0^{(t+1)e} = \boldsymbol{z}_0^{td}$.

The specific workflow of the Lmser block is as follows: when a sketch $\boldsymbol{X}$ is fed to the Lmser block, the neurons of the encoder are activated layer by layer. Then, the embedding $\boldsymbol{z}_L^{0e}$ obtained from the top-most layer of the encoder are sent to the decoder. The neurons in each layer of the decoder are initialized by receiving both the information transmitted by the forward connection and the signals from the previous layer of the decoder, e.g. $\boldsymbol{z}_1^{0d} = \alpha g^{(2)}(\boldsymbol{z}_2^{0d}) + (1-\alpha)\boldsymbol{z}_1^{0e}$. The decoder output, $\boldsymbol{z}_0^{0d}$, is used as the input $\boldsymbol{z}_0^{1e}$ to the encoder in the next loop. Thereafter, the updates of the neurons in the encoder are also influenced by the information delivered by the feedback connections, e.g. $\boldsymbol{z}_2^{1e} = \alpha f^2(\boldsymbol{z}_1^{1e}) + (1-\alpha)\boldsymbol{z}_2^{0d}$. The neurons in the Lmser are dynamic updated layer by layer through a large loop repeatedly, see Fig. 2 of CLmser (Huang et al., 2020b) for more details.

After $T$ loops, the top-most and bottom-most layers of the Lmser block output the feature $\boldsymbol{y}$ and the reconstructed sketch $\hat{\boldsymbol{X}}$, respectively. Theoretically, the dynamic process above can eventually reach an equilibrium state (Xu, 1993). In practice, the choice of $T$ is a trade-off between performance and consumption. More iterations allow the feature maps to be fully fused and facilitate the output of stable features, but also require the involvement of more computational resources. In this paper, we set $T$ to 2 by default. It is worth noting that when $T = 0$, Lmser will degenerate into a U-Net-like network architecture.

## 3.3 TRAINING LMSER-PIX2SEQ

Lmser-pix2seq has a multi-modal output property whose main purpose is to make the recreated vector sketch $\hat{\boldsymbol{S}}$ as high-quality as possible while maintaining global semantic with the original full sketch $\tilde{\boldsymbol{S}}$. To obtain a better reconstruction in multi-class sketch, we remove the Kullback-Leibler (KL) Divergence Loss term in VAE as existing work (Chen et al., 2017; Su et al., 2020; Qi et al., 2021; 2022). The loss function consists of two parts: the raster image reconstruction loss $\mathcal{L}_{CNN}$ and the sequence reconstruction loss $\mathcal{L}_{RNN}$. As commonly image generative tasks, $\mathcal{L}_{CNN}$ adopts the $l_2$ loss. Following sketch-rnn (Ha & Eck, 2018), the sequence modal's reconstruction loss $\mathcal{L}_{RNN}$ is to minimize the negative log-likelihood of the generated probability distribution (Qi et al., 2021). To sum up, our objective is to minimize

$$
\begin{aligned}
\mathcal{L}_{total} &= w_c \mathcal{L}_{CNN} + w_r \mathcal{L}_{RNN} \\
&= \frac{1}{2} w_c \mathbb{E}(\|\tilde{\boldsymbol{X}} - \hat{\boldsymbol{X}}\|_2^2) - w_r \mathbb{E}_{q_\phi(\boldsymbol{m}|\boldsymbol{X})}[\log p_{\boldsymbol{\theta}}(\hat{\boldsymbol{S}}|\boldsymbol{m})],
\end{aligned}
\tag{2}
$$

where $\tilde{\boldsymbol{X}}$ and $\hat{\boldsymbol{X}}$ denote the full sketch and the complete sketch reconstructed by Lmser block, respectively. $\mathcal{L}_{CNN}$ and $\mathcal{L}_{RNN}$ are weighted by hyper-parameters $w_c$ and $w_r$ correspondingly.

## 4 EXPERIMENT

### 4.1 PREPARATION

**Dataset.** Our experiments are conducted on two datasets, a 17-category dataset 1 (DS1) [1] from SketchHealer-1.0 (Su et al., 2020) and a 5-category dataset 2 (DS2) [2] from RPCL-pix2seq (Zang et al., 2021). DS1 and DS2 are both from the large-scale dataset Quickdraw (Ha & Eck, 2018). Each category contains 70000 sketches for training and 2500 sketches for testing.

Our corruption method is generally consistent with SketchHealer-1.0 (Su et al., 2020). The sketch is drawn with a $640 \times 640$ canvas, and $M$ patches of $128 \times 128$ are taken from the canvas according to the order of the stroke points, then some of the patches are removed with a probability $p_{mask}$. In the end, we resize the corrupted canvas into a $128 \times 128$ image as the corrupted sketch. SketchHealer-1.0's mask method (Su et al., 2020) has an information leakage problem. There are two patches A and B with overlapping parts, A is cropped before B, and B is selected to be removed, at this time, A still contains some information from the removed B. Our solution to the information leakage problem in SketchHealer-1.0 (Su et al., 2020) is to first mask all the information on the canvas corresponding to the location of the patches that need to be removed, and then crop the individual patches.

**Evaluation Metrics.** $Rec$ and $Ret$ (Zang et al., 2021) are evaluated as recreated sketch metrics. $Rec$ indicates whether the recreated sketch $\hat{S}$ and its corresponding sketch $\tilde{S}$ belong to the same category, e.g. $top-1$ recognition accuracy. We pre-train two sketch a net classifiers (Yu et al., 2015) for computing $Rec$ in two datasets, respectively. $Ret$ is a measure of the semantic similarity between the recreated sketch $\hat{S}$ and its corresponding full sketch $\tilde{S}$. Specifically, for different models, full sketch $\tilde{S}$ and recreated sketch $\hat{S}$ are rasterized and fed to their own encoder to obtain vectors $\tilde{\mu}$ and $\hat{\mu}$. We use $\hat{\mu}$ to retrieve its corresponding vector $\tilde{\mu}$, and $Ret$ is the success rate of retrieval. We use $Ret@k$ to represent $top-k$ retrieval accuracy. Our $Ret$ is completely different from SketchHealer-1.0 (Su et al., 2020), SketchHealer-2.0 (Qi et al., 2022) and SketchLattice (Qi et al., 2021), whose retrieval is a category-level retrieval and not an instance-level retrieval. Our $Ret$ is more responsive to the semantic consistency of the recreated sketch and the full sketch.

**Baseline.** We compare our Lmser-pix2seq with five baseline models in the sketch healing task. These models include sketch-pix2seq (Chen et al., 2017) and RPCL-pix2seq (Zang et al., 2021), which are proposed for sketch generation. Additional models are SketchHealer-1.0 (Su et al., 2020), SketchHealer-2.0 (Qi et al., 2022), and SketchLattice (Qi et al., 2021), which are designed for sketch healing. Among these models, only RPCL-pix2seq has constraints on the latent space.

We train SketchLattice (Qi et al., 2021) and SketchHealer-1.0 (Su et al., 2020) using the official open source code. Meanwhile, we re-implement RPCL-pix2seq by pytorch (Paszke et al., 2017). The code of SketchHealer-2.0 (Qi et al., 2022) is not open source yet, while the original paper lacks the specific implementation of gradient back-propagation after rasterization. Therefore, we do our best to reproduce SketchHealer-2.0. We first pre-train multi-category sketch-rnn (Ha & Eck, 2018) networks for each datasets and freeze the networks' parameters. When training SketchHealer-2.0, we choose the Gaussian component and pen state with the highest probability of the LSTM output to generate a five-tuple vector sequence $\hat{S}$ to ensure that the gradients can be back-propagated. Eventually, $\hat{S}$ and $\tilde{S}$ are simultaneously fed into the pre-trained sketch-rnn's encoder to get the corresponding latent code, which are used to calculate the perceptual loss. Since the masking process is performed on the canvas, we do not include networks like sketch-rnn, which use sequences as input, as the baseline models.

**Implement Details.** The Adam optimizer (Kingma & Ba, 2014) is applied to our Lmser-pix2seq with parameters $\beta_1 = 0.9$, $\beta_2 = 0.999$, $\epsilon = 10^{-8}$, and the learning rate of the network starts from $10^{-3}$ with a decay rate of 0.999 for every iteration. We randomly mask $p_{mask} = 10\%$ of the patches to obtain the corrupted sketch for model training, and different proportions for model testing. The weights of the loss function are $w_c = 0.5$ and $w_r = 1$. The dimension of the latent code $m$ is $N_z = 128$. All models is trained on a single NVIDIA RTX 2080Ti GPU with 150000 iterations.

---

[1]airplane, angel, alarm clock, apple, butterfly, belt, bus, cake, cat, clock, eye, fish, pig, sheep, spider, umbrella, The Great Wall of China. These categories are common in life and the instances in the categories are globally similar in appearance.

[2]bee, bus, flower, giraffe, pig. These classes have multi-style characteristics and are more challenging for sketch healing.

4.2 RESULTS

**Quantitative Results**. Table 1 reports the retrieval performance on DS1. The proposed Lmser-pix2seq learns stable representations which lead to outstanding performance on $Ret$. When the sketch is unmasked, the success rate of Lmser-pix2seq retrieval performs significantly superior to other methods. The top-1 $Ret$ of our method is 22.91% higher than SketchHealer-1.0 (Su et al., 2020). The advantage of the Lmser-pix2seq becomes more apparent when the proportion of sketches masked gradually increases. When $p_{mask}$=50%, the top1 $Ret$ of Lmser-pix2seq improved by 33.42% compared with SketchHealer-1.0. On the multi-style DS2, our method still performs well, see Table 2. The $Rec$ metrics of Lmser-pix2seq are overall at the same level as the state-of-the-art methods SketchHealer-1.0 (Su et al., 2020) and SketchHealer-2.0 (Qi et al., 2022). The above results show that our model satisfies the need for both high-quality and semantic preservation. Observe the performance of other models. RPCL-pix2seq (Zang et al., 2021), as a latent space constrained model, focuses on the controllability of sketch generation rather than sketch healing, which makes it more sensitive to masking rates than sketch-pix2seq (with KL term removed) (Chen et al., 2017). The lightweight SketchLattice (Qi et al., 2021) takes coordinates as input, making it difficult to utilize sufficient information as input. Besides, when there are noisy strokes in the sketch, inappropriate coordinates may be treated as nodes by SketchLattice and affect the quality of sketch generation. Another reason is that we are performing the lattice operation on the corrupted sketch instead of discarding several nodes based on probability as in (Qi et al., 2021), which causes more information to be removed. Naturally, SketchLattice*, which uses the original masking method, has improved both $Rec$ and $Ret$ SketchHealer-1.0 (Su et al., 2020) and SketchHealer-2.0 (Qi et al., 2022) produce higher quality generation of corrupted sketches, as reflected in the $Rec$ metric. SketchHealer-2.0 shows an improvement on $Rec$ compared with SketchHealer-1.0, which benefits from the perceptual loss to promote global semantic preservation. However, the $Ret$ of SketchHealer-2.0 do not perform consistently between DS1 and DS2. This may be due to the fact that we reproduce the perceptual loss calculation by sketch-rnn (Ha & Eck, 2018), which is not designed for multi-category sketch generation.

**Qualitative Results**. In Fig. 2, we present the sketch healing results for $p_{mask} = 10\%$, $p_{mask} = 30\%$ and $p_{mask} = 50\%$. With less loss of sketch information ($p_{mask} = 10\%$), the healing results for all models are relatively reasonable. However, as more strokes are covered, not all methods are effective. When $p_{mask} = 50\%$, some key details in the sketches are lost, e.g., the head of the sheep is almost completely lost. Lmser-pix2seq successfully recreate the sheep by the limited stroke information of the head, while other methods can only produce a rough body. Observe the airplane in Fig. 2. When $p_{mask} = 30\%$ and $p_{mask} = 50\%$, the shape of the fuselage in most of the generated sketches does not match the shape in the original sketch. However, the sketch recreated by Lmser-pix2seq and SketchHealer-2.0 (Qi et al., 2022) maintains the structure of the fuselage

Table 1: Sketch healing retrieval performance $Ret$ (%) ↑ on DS1.

| Model | $p_{mask}$ | $Ret$ @10 | $Ret$ @30 | $Ret$ @50 | $p_{mask}$ | $Ret$ @10 | $Ret$ @30 | $Ret$ @50 |
|---|---|---|---|---|---|---|---|---|
| sketch-pix2seq | 0% | 31.88 | 48.35 | 58.33 | 30% | 22.08 | 37.76 | 48.65 |
| (Chen et al., 2017) | 10% | 28.74 | 45.13 | 55.72 | 50% | 15.38 | 28.81 | 39.78 |
| RPCL-pix2seq | 0% | 32.15 | 47.84 | 57.71 | 30% | 17.29 | 29.76 | 40.15 |
| (Zang et al., 2021) | 10% | 26.69 | 41.54 | 52.21 | 50% | 10.26 | 19.51 | 28.48 |
| SketchLattice | 0% | 0.10 | 0.70 | 2.27 | 30% | 0.07 | 0.42 | 1.57 |
| (Qi et al., 2021) | 10% | 0.09 | 0.58 | 2.07 | 50% | 0.04 | 0.29 | 1.19 |
| SketchLattice* | 0% | 0.10 | 0.70 | 2.27 | 30% | 0.10 | 0.60 | 2.33 |
| (Qi et al., 2021) | 10% | 0.14 | 0.72 | 2.47 | 50% | 0.06 | 0.41 | 1.54 |
| SketchHealer-1.0 | 0% | 54.29 | 70.42 | 79.22 | 30% | 34.38 | 52.05 | 63.96 |
| (Su et al., 2020) | 10% | 47.07 | 63.49 | 73.40 | 50% | 21.47 | 38.21 | 51.51 |
| SketchHealer-2.0 | 0% | 50.83 | 66.50 | 75.31 | 30% | 34.56 | 51.76 | 63.20 |
| (Qi et al., 2022) | 10% | 43.20 | 59.58 | 69.58 | 50% | 19.15 | 35.48 | 49.36 |
| Lmser-pix2seq | 0% | **77.20** | **84.82** | **88.71** | 30% | **65.83** | **76.78** | **82.59** |
| | 10% | **73.92** | **82.59** | **87.01** | 50% | **54.89** | **64.70** | **75.36** |

Table 2: Sketch healing retrieval performance $Ret$ (%) $\uparrow$ on DS2.

| Model | $p_{mask}$ | $Ret$ @10 | $Ret$ @30 | $Ret$ @50 | $p_{mask}$ | $Ret$ @10 | $Ret$ @30 | $Ret$ @50 |
|---|---|---|---|---|---|---|---|---|
| sketch-pix2seq | 0% | 30.29 | 52.90 | 66.25 | 30% | 17.06 | 36.10 | 52.98 |
| (Chen et al., 2017) | 10% | 25.70 | 48.04 | 63.31 | 50% | 10.83 | 25.75 | 41.11 |
| RPCL-pix2seq | 0% | 34.74 | 56.49 | 71.50 | 30% | 12.38 | 28.56 | 45.14 |
| (Zang et al., 2021) | 10% | 24.77 | 46.22 | 63.83 | 50% | 6.02 | 15.90 | 29.77 |
| SketchLattice | 0% | 0.32 | 1.97 | 7.03 | 30% | 0.15 | 1.12 | 4.31 |
| (Qi et al., 2021) | 10% | 0.33 | 1.95 | 6.13 | 50% | 0.14 | 1.08 | 3.51 |
| SketchLattice* | 0% | 0.32 | 1.97 | 7.03 | 30% | 0.33 | 2.07 | 6.58 |
| (Qi et al., 2021) | 10% | 0.41 | 2.10 | 6.76 | 50% | 0.31 | 1.87 | 6.26 |
| SketchHealer-1.0 | 0% | 56.99 | 72.29 | 80.48 | 30% | 30.06 | 48.84 | 62.54 |
| (Su et al., 2020) | 10% | 47.34 | 64.70 | 74.80 | 50% | 16.22 | 33.29 | 48.43 |
| SketchHealer-2.0 | 0% | 59.98 | 75.90 | 84.14 | 30% | 32.98 | 52.36 | 66.50 |
| (Qi et al., 2022) | 10% | 49.54 | 67.99 | 77.99 | 50% | 18.14 | 35.71 | 51.03 |
| Lmser-pix2seq | 0% | **81.25** | **86.96** | **90.66** | 30% | **65.94** | **77.65** | **84.22** |
| | 10% | **78.06** | **83.58** | **88.16** | 50% | **52.86** | **67.94** | **77.06** |

Table 3: Sketch healing recognition performance $Rec$ (%) $\uparrow$.

| Dataset | Model | $p_{mask}$ 0% | 10% | 30% | 50% |
|---|---|---|---|---|---|
| DS1 | sketch-pix2seq (Chen et al., 2017) | 66.99 | 66.02 | 60.68 | 53.40 |
| | RPCL-pix2seq (Zang et al., 2021) | 69.86 | 66.15 | 55.19 | 44.33 |
| | SketchLattice (Qi et al., 2021) | 48.88 | 46.57 | 37.87 | 27.91 |
| | SketchLattice* (Qi et al., 2021) | 48.88 | 49.15 | 45.85 | 31.16 |
| | SketchHealer-1.0 (Su et al., 2020) | 76.76 | 75.05 | 70.47 | 62.86 |
| | SketchHealer-2.0 (Qi et al., 2022) | 77.48 | 75.76 | 71.93 | 64.56 |
| | Lmser-pix2seq | **78.48** | **77.57** | **74.73** | **70.12** |
| DS2 | sketch-pix2seq (Chen et al., 2017) | 88.36 | 87.04 | 82.84 | 75.88 |
| | RPCL-pix2seq (Zang et al., 2021) | 90.66 | 87.32 | 76.07 | 63.15 |
| | SketchLattice (Qi et al., 2021) | 77.54 | 76.18 | 70.42 | 62.38 |
| | SketchLattice* (Qi et al., 2021) | 77.54 | 77.81 | 77.10 | 72.98 |
| | SketchHealer-1.0 (Su et al., 2020) | 90.93 | 90.50 | 88.80 | 83.15 |
| | SketchHealer-2.0 (Qi et al., 2022) | 92.10 | 91.74 | 90.01 | 84.23 |
| | Lmser-pix2seq | **92.42** | **91.78** | **90.49** | **88.37** |

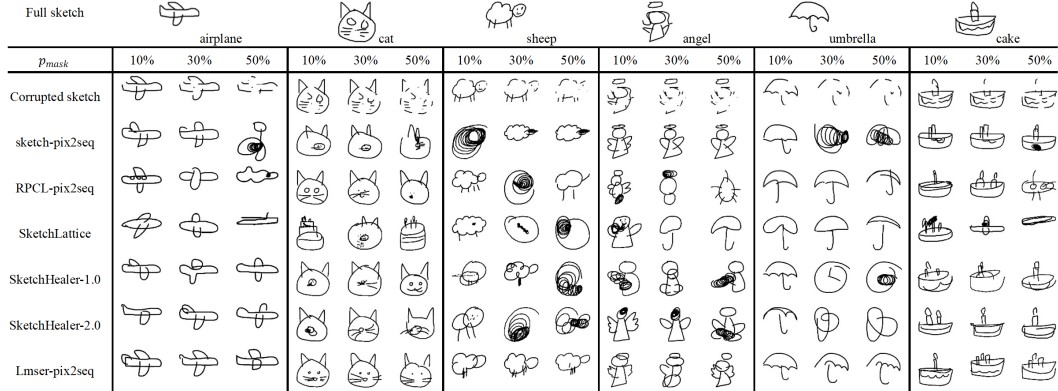

Figure 2: The healing results with masking probabilities of 10%, 30%, and 50%.

better, especially the tail section. Fig. 3 shows the sketch healing results for different masking areas at $p_{mask} = 30\%$. Compared with the state-of-the-art methods, the healing results of the proposed Lmser-pix2seq do not differ significantly with the change of the masking location.

Figure 3: Qualitative comparisons between the proposed Lmser-pix2seq and other state of-the-art methods. $p_{mask} = 30\%$ throughout.

## 4.3 ABLATION STUDY

In this subsection, we examine the effectiveness of the skip connections, the number of iterations $T$, and the image reconstruction. Our ablation experiments are conducted on the larger DS1. We design five models, as shown in Table 4, of which Lmser-pix2seq (T=2) is the Lmser-pix2seq model above. Table 5 and Table 6 shows the $Ret$ and $Rec$ results of the ablation experiments, respectively.

**CNN Decoder.** CAE-pix2seq with CNN decoder generates more recognizable sketches (higher $Rec$) compared with CE-pix2seq. This is because the CNN decoder captures the overall structure of a certain class of sketches by reconstructing the images and guides the CNN encoder to extract more accurate features (Zang et al., 2021). However, the deep network makes it difficult to efficiently transfer details in sketches from the shallow encoder layers to the deep decoder layers, which results in poor $Ret$ performance of the CAE-pix2seq.

Table 4: Models for ablation study.

| Model | CNN encoder | CNN decoder | Skip connection | $T$ |
|---|---|---|---|---|
| CE-pix2seq | Yes | No | No | - |
| CAE-pix2seq | Yes | Yes | No | - |
| Lmser-pix2seq (T=1) | Yes | Yes | Yes | 1 |
| Lmser-pix2seq (T=2) | Yes | Yes | Yes | 2 |
| Lmser-pix2seq (T=4) | Yes | Yes | Yes | 4 |

Table 5: Sketch healing retrieval performance $Ret$ (%) ↑ on DS1 for ablation study.

| Model | $p_{mask}$ | $Ret$ @10 | $Ret$ @30 | $Ret$ @50 | $p_{mask}$ | $Ret$ @10 | $Ret$ @30 | $Ret$ @50 |
|---|---|---|---|---|---|---|---|---|
| CE-pix2seq | 0% | 55.15 | 73.50 | 82.01 | 30% | 39.55 | 60.90 | 73.57 |
| | 10% | 50.13 | 69.85 | 80.07 | 50% | 29.84 | 50.66 | 65.17 |
| CAE-pix2seq | 0% | 48.38 | 64.21 | 73.20 | 30% | 34.97 | 51.70 | 62.82 |
| | 10% | 43.72 | 60.08 | 69.91 | 50% | 26.01 | 41.87 | 54.09 |
| Lmser-pix2seq (T=1) | 0% | 72.40 | 80.75 | 85.84 | 30% | 58.70 | 70.06 | 77.24 |
| | 10% | 68.20 | 77.81 | 83.52 | 50% | 47.48 | 60.24 | 68.51 |
| Lmser-pix2seq (T=2) | 0% | 77.20 | 84.82 | 88.71 | 30% | **65.83** | **76.78** | **82.59** |
| | 10% | 73.92 | **82.59** | **87.01** | 50% | **54.89** | 64.70 | **75.36** |
| Lmser-pix2seq (T=4) | 0% | **77.70** | **84.87** | **88.75** | 30% | 65.61 | 75.80 | 81.71 |
| | 10% | **74.58** | 82.57 | 86.97 | 50% | 54.07 | **66.10** | 73.59 |

Table 6: Sketch healing recognition performance $Rec$ (%) ↑ on DS1 for ablation study.

| Dataset | Model | $p_{mask}$ | | | |
|---|---|---|---|---|---|
| | | 0% | 10% | 30% | 50% |
| DS1 | CE-pix2seq | 67.77 | 67.09 | 64.42 | 60.73 |
| | CAE-pix2seq | 72.38 | 70.66 | 66.60 | 61.60 |
| | Lmser-pix2seq (T=1) | 76.82 | 75.44 | 71.92 | 66.49 |
| | Lmser-pix2seq (T=2) | 78.48 | 77.57 | 74.73 | **70.12** |
| | Lmser-pix2seq (T=4) | **79.73** | **78.48** | **75.38** | 69.74 |

**Skip Connection.** The performance of Lmser-pix2seq (T=1) is significantly improved over CAE-pix2seq by introducing skip connections, especially on $Ret$. The main reason is that while the forward connections assist the decoder in reconstructing the sketch, the feedback connections transmit the reconstruction information to the encoder for feature map fusion, capturing the exact feature of the instance.

**Number Of Iterations.** The experimental results show that the performance of Lmser-pix2seq with $T = 2$ is obviously superior to that with $T = 1$. This indicates that one cycle does not allow the feature maps in the Lmser block to be fully fused, and the feature extracted from the sketch is not stable. When T=4, the $Rec$ metric of Lmser-pix2seq improves overall, compared with the Lmser-pix2seq (T=2). However, the benefit of more iterations with resource consumption is an issue that should be considered.

## 5 CONCLUSION

In this paper, we present Lmser-pix2seq to learn stable sketch representations for sketch healing. The bi-directional skip connections in our Lmser block facilitate the fusion of the feature maps between the encoder and decoder, thus capturing the stable features of the corrupted sketch. Benefiting from the stable features, the Lmser-pix2seq can recreate sketches with high-quality and preserve their global semantics. Experiments show that our approach outperforms the state-of-the-art models in the sketch healing task.

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

# A THE DIFFERENCES BETWEEN SKETCH-PIX2SEQ, RPCL-PIX2SEQ AND LMSER-PIX2SEQ

sketch-pix2seq, RPCL-pix2seq and Lmser-pix2seq differ in the problems they address and in the design of the network architecture. The sketch-pix2seq is concerned with category-level generation, which leads to a high Rec score. The main function of RPCL-pix2seq is to generate sketches according to given categories or styles. Lmser-pix2seq focuses on capturing features at the sketch instance-level, with the strength of high Ret score. For model design, sketch-pix2seq uses a CNN encoder to extract features and an RNN decoder to output recreated sketches. RPCL-pix2seq introduces a CNN decoder on top of sketch-pix2seq and assumes the latent space follows a Gaussian mixture model (GMM). The CNN decoder in RPCL-pix2seq plays a role of regularization and helps the formation of feature concepts. The network architecture of RPCL-pix2seq is similar to the CAE-pix2seq model in section 4.3. Lmser-pix2seq constructs bi-directional skip connections between the CNN encoder and CNN decoder. This allows both the CNN encoder and the CNN decoder to be involved in feature extraction and image reconstruction. Loops obtained based on bi-directional skip connections are also discussed. Experimental results show that two and more cycles are effective in capturing sketch features.

# B SKETCH HEALING RESULTS FOR IRREGULAR HOLE MASKING

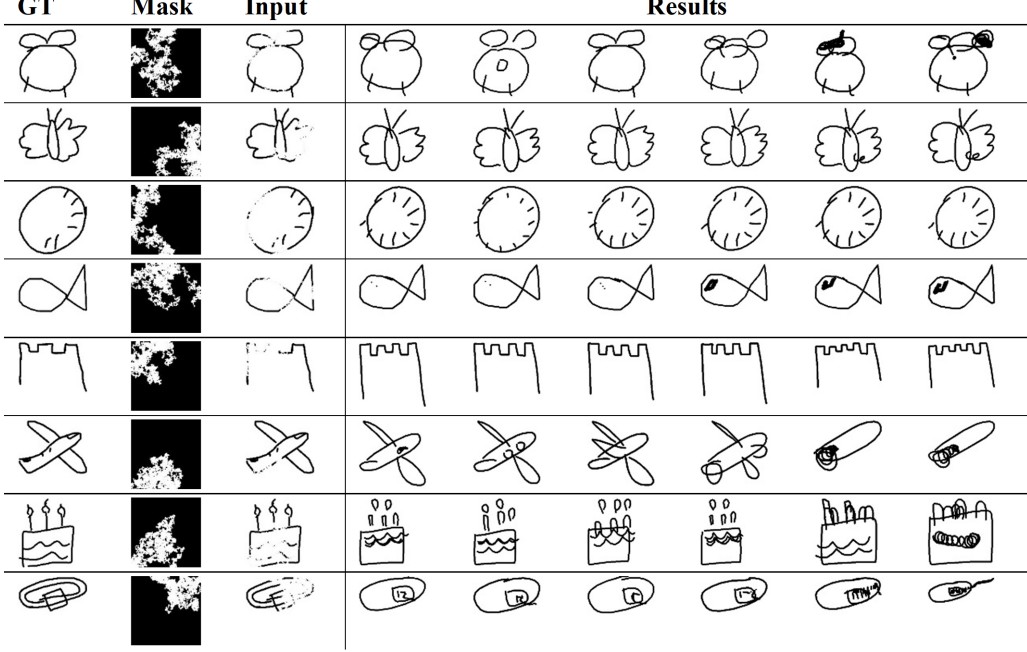

Figure 4: Sketch healing results for irregular hole masking.The left three columns in the figure are ground truth sketches, masks and masked sketches. The right six columns in the figure are the sketches recreated by Lmser-pix2seq.

# C RETRIEVAL METRIC

Fig. 5 shows two retrieval methods, and the metric used in this paper correspond to (a) in the figure. Compared with the retrieval metric in SketchHealer-1.0 (Su et al., 2020), $Ret$ is more challenging. For each query, $Ret$ allows only one correct result, while hundreds of answers exist for the other method.

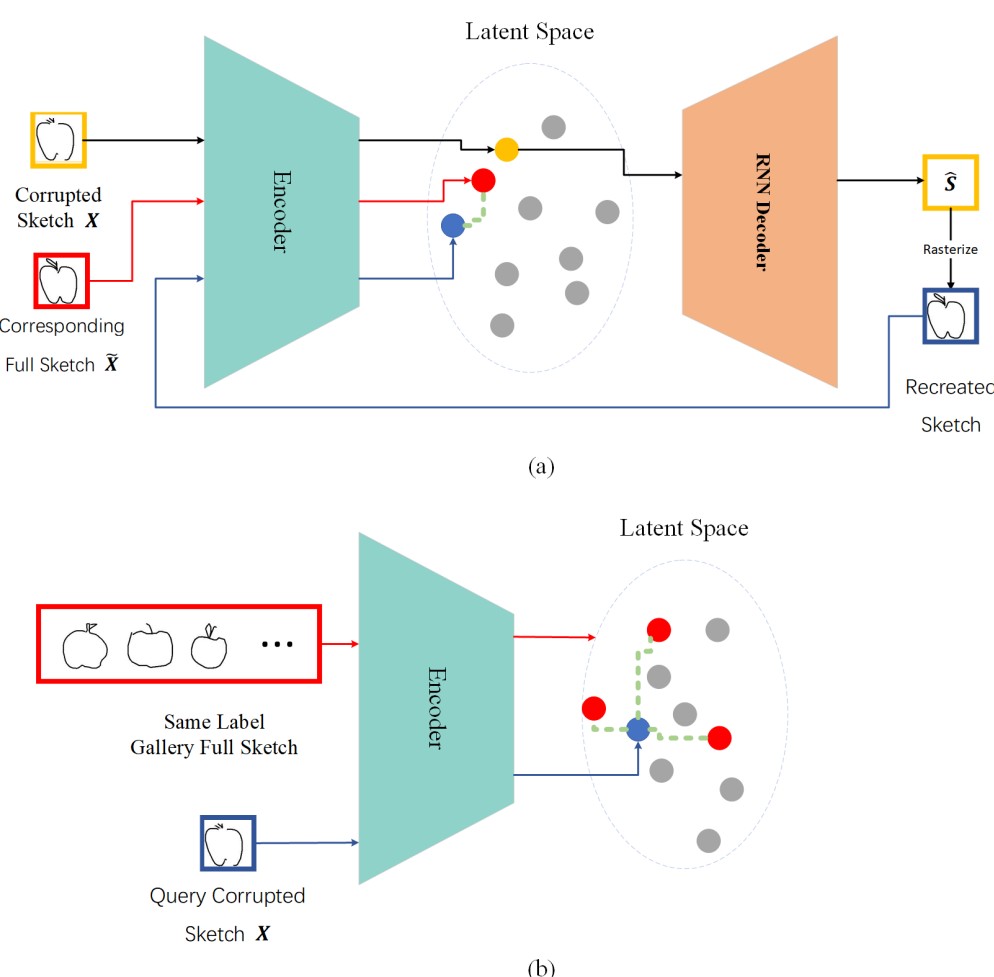

(a)

(b)

Figure 5: Two types of retrieval metrics. The retrieval is performed in the latent space. Blue circles represent queries and red circles represent samples with correct retrieval results. (a) The retrieval method from RPCL-pix2seq (Zang et al., 2021), i. e. $Ret$, is an instance-level approach. (b) The retrieval method from SketchHealer-1.0 (Su et al., 2020) is a category-level approach.

# D    LMSER-PIX2SEQ ONLY TRAINED WITH CNNS

When Lmser-pix2seq only trained with CNNs, sketch healing is then turned into image inpainting. Table 7 and Table 8 report the relevant metrics. Fig. 6 shows the results of some sketch reconstructions. Although these images maintain the overall structure of the sketch well, some of the results are blurry and incomplete. This is why an RNN decoder is needed to generate the sequences.

Table 7: Sketch image inpainting recognition performance $Rec$ (%) $\uparrow$ .

| Dataset | Model | $p_{mask}$ | | | |
| --- | --- | --- | --- | --- | --- |
| | | 0% | 10% | 30% | 50% |
| DS1 | CNNs | 89.28 | 87.19 | 80.56 | 71.07 |
| DS2 | CNNS | 96.10 | 95.60 | 94.33 | 92.12 |

Table 8: Sketch image inpainting retrieval performance $Ret$ (%) ↑.

| Model | $p_{mask}$ | $Ret$ @10 | $Ret$ @30 | $Ret$ @50 | $p_{mask}$ | $Ret$ @10 | $Ret$ @30 | $Ret$ @50 |
|---|---|---|---|---|---|---|---|---|
| CNNs DS1 | 0% | 98.81 | 99.38 | 99.55% | 63.45 | 73.78 | 79.76 | |
| | 10% | 87.33 | 92.00 | 94.17 | 50% | 44.30 | 56.37 | 64.65 |
| CNNs DS2 | 0% | 97.73 | 98.52 | 98.92% | 55.77 | 67.81 | 75.09 | |
| | 10% | 82.74 | 88.46 | 91.22 | 50% | 35.48 | 48.65 | 58.20 |

Figure 6: When only CNNs is trained, some results of CNN decoder output.

