# OpenReview forum: "Lmser-pix2seq: Learning Stable Sketch Representations For Sketch Healing"
_ICLR.cc/2023/Conference — Submitted to ICLR 2023_

### Official Review · Reviewer_po7R · 2022-10-23

**Confidence:** 5
**Correctness:** 3
**Technical Novelty And Significance:** 4
**Empirical Novelty And Significance:** 2
**Recommendation:** 8

**Clarity, Quality, Novelty And Reproducibility:**

Writing can be improved specifically on the Introduction. More notes will follow at the end of this comment.

I believe this is a paper with a single contribution, which is the application of Lmser to the task of Sketch Healing. It is focused and delivers on the experimental results; that said I find it is not necessary to break this single contribution into three which are each derived from the previous one.

In the related work section, I find it odd that the authors define reconstruction as “conditional generation” and class-conditional generation as “unconditional generation”. The task of sketch healing is similar to the one of reconstructing images from noisy inputs and unless there is a pattern in the sketch literature for this naming convention it would be better to follow the general computer vision conventions.

The authors mention the LSTM model is referenced from Sketch-RNN, does that include the GMM model at the output?

In section 3.1, the manuscript mentions that corrupted sketches are obtained by “randomly rounding off a certain percentage of strokes”, which would make corrupt sketches always be the same as unfinished sketches. It is not clear however when this corruption technique is used since in section 4.1 another common method is described to be applied for the experiments. Was the first method used for training the model and the second used for testing?

The naming and explanations of the metrics Rec and Ret could be amended to better explain what are the actual metrics used. Is Rec simple top-1 accuracy? Is Ret precision@K? The short names can be used on the tables but please make meaning explicit within the text.

Extra notes:
- Sec 1. “due to the difficulty in learning” can be removed from the abstract and introduction
- Sec 1. “it is not always available” needs a better connective “but this information” would work
- Sec 1. “SketchLattice is a lightweight model which limits its performance” is not always a true statement for lightweight models.
- Sec 1. “Different to […] we expect the network to learn stable sketch representations”. This motivator does not explain why a cnn would be better than a gnn at learning stable representations
- Sec 1. “and even more” should be “and worse” or similar
- Sec 1. “oin” should be “in”
- Sec 3.3. “raseter”


**Strength And Weaknesses:**

The strength is in the core contribution of the paper: the choice of using the Lmser blocks is well motivated by the need for stable representations for complete and corrupted sketches; the idea is new to the sketch literature; the empirical results show the effectiveness of using this new representation combined with the traditional recurrent decoder.

For weaknesses, I’ve found the comparison agains the SketchLattice baseline to be flawed. The input requirements for that model should have been followed to allow for fair comparison. Furthermore, other competing models such as SketchHealer-2.0 have presented results aligned with the original SketchLattice papers in their comparisons. I am also curious as to how well the model would work when only trained with the CNNs, without the vector decoder. I understand the RNN decoder to be essential for the output to not be blurry, but is it also essential for obtaining a stable representation?


**Summary Of The Paper:**

The authors propose a new solution for the task of Sketch healing, where corrupted or unfinished sketches are reconstructed without the corruption. This task is commonly accomplished through representation learning. The manuscript describes a method focused on ‘stable representations’, where complete and corrupted sketches should have similar representations; this is a concept not dissimilar to the principles of representation learning in self-supervision models. To achieve such stability the model is composed of a CNN-based autoencoder with least mean square error reconstruction (Lmser) blocks that reconstruction raster sketches plus an RNN-based decoder that reconstructs vector sketches. The manuscript also has empirical evidence supporting this new model as the State-of-the-Art on sketch healing.


**Summary Of The Review:**

The paper applies the existing Lmser AEs method to the task of learning stable representations for sketches, with the final goal of performing sketch healing. The authors combine the Lmser AE with the usual recurrent decoder for vector sketch generation. While there are flaws with one of the baseline comparisons and the ablations could be more thorough at figuring out why this model works, the performance gains are significant and I believe this model would be a good addition to the sketch community’ literature.

---

> ### Author Response · Authors · 2022-11-16
> **Author Responses to Reviewer po7R (1/2)**
>
> Thanks for your positive and constructive feedback.
>
> $\textbf{Q1: For weaknesses, I’ve found the comparison against the SketchLattice baseline to be flawed. }$$\textbf{The input requirements for that model should have been followed to allow for a fair comparison.}$
>
> A1: Our comparison is fair. The reason we modified the input to SketchLattice is that we wanted the corrupted sketches to be completely consistent across different models when performing the test. We recalculated the Ret of SketchLattice and report the metrics for the original SketchLattice (SketchLattice* for short) as follows：
>
> Since SketchLattice's masking method will lose more information, the Rec and Ret of SketchLattice are lower than SketchLattice *.
>
> Rec top-1 (%):
>
> DS 1
>
>     p_mask		 0.0	0.1	0.3	0.5
>
>     SketchLattice	48.88  46.57 	37.87 	27.91
>
>     SketchLattice*	48.88  49.15 	45.85 	31.16
>
> DS 2
>
>     p_mask		 0.0	0.1	0.3	0.5
>
>     SketchLattice	77.54  76.18   70.42  62.38
>
>      SketchLattice*	77.53  77.81   77.10  72.98
>
> Ret (%):
> DS1
>
>                       p_mask  	Ret@1	Ret@10	Ret@50
>
> 	SketchLattice	  0.0		 0.10 	 0.70 	 2.27
> 				      0.1            0.09 	 0.58 	 2.07
> 				      0.3 	         0.07 	 0.42 	 1.57
> 				      0.5 	         0.04 	 0.29 	 1.19
>
> 	SketchLattice*   0.0		 0.10 	 0.70 	 2.27
> 				     0.1 	         0.14 	 0.72 	 2.47
> 				     0.3 	         0.10 	 0.60 	 2.33
> 				     0.5 	         0.06 	 0.41 	 1.54
>
> DS2
>
>                       p_mask  	Ret@1	Ret@10	Ret@50
>
>       SketchLattice	0.0		0.32 	 1.97 	 7.03
> 				    0.1 	        0.33 	 1.95 	 6.13
> 				    0.3 	        0.15 	 1.12 	 4.31
> 				    0.5 	        0.14 	 1.08 	 3.51
>
> 	SketchLattice* 0.0		0.32 	 1.97 	 7.03
> 				   0.1 	        0.33 	 2.10 	 6.76
> 				   0.3 	        0.33 	 2.07 	 6.58
> 				   0.5 	        0.31 	 1.87 	 6.26
>
> $\textbf{Q2: When only trained with the CNNs, without the vector decoder.}$
>
> A2: When only trained with the CNNs, we take the output of the CNN decoder to calculate Rec and Ret. The results are as follow:
> Rec top-1 (%):
>
> DS 1
>
>     p_mask		 0.0	0.1	0.3	0.5
>
>      CNNs		89.28 	87.19 	80.56 	71.07
>
> DS 2
>
>     p_mask		 0.0	0.1	0.3	0.5
>
>      CNNs		96.10 	95.60 	94.33 	92.12
>
> Ret (%):
>
> DS1
>
>             p_mask  	Ret@1	Ret@10	Ret@50
>
> 	CNNs	0.0		 97.73 	 98.52   98.92
> 			0.1 	         82.74 	 88.46 	 91.22
> 			0.3 	         55.77 	 67.81 	 75.09
> 			0.5 	         35.48 	 48.65 	 58.20
>
> DS2
>
>             p_mask  	Ret@1	Ret@10	Ret@50
>
> 	CNNs	0.0		 98.81 	 99.38 	 99.55
> 			0.1 	         87.33 	 92.00 	 94.17
> 			0.3 	         63.45 	 73.78	 79.76
> 			0.5 	         44.30	 56.37	 64.65
>
> It is only trained with CNNs that implement the image reconstruction task rather than sketch generation.
>
> $\textbf{Q3: The single contribution.}$
>
> A3: We have merged our contributions into one article, as follows:
> “In summary, our contribution is that we propose Lmser-pix2seq to learn stable sketch representations for sketch healing. The bi-directional skip connections in our Lmser blocks allow the feature maps from the encoder and decoder to be sufficiently fused to facilitate the extraction of sketch features. Experimental results show that our Lmser-pix2seq outperforms the state-of-the-art methods, especially when the sketches are heavily masked or corrupted.”
>
> $\textbf{Q4: In the related work section, I find it odd that the authors define reconstruction as “conditional generation” }$$\textbf{and class-conditional generation as “unconditional generation”.}$
>
> A4: We have changed this expression to avoid confusion, as follows:
> “Research related to sketch generation with deep learning methods has been developing rapidly in recent years. An interesting work on sketch generation is that the neural network imitates humans to draw the vector sketch stroke by stroke.”
> The concept of conditional and unconditional generation in previous related work comes from sketch-rnn [3], however, our formulation is not precise enough.
>
> $\textbf{Q5: The authors mention the LSTM model is referenced from Sketch-RNN, does that include the GMM model at the output?}$
>
> A5: Our method includes the GMM model at the output. We supplemented the first paragraph of section 3.1 as follows:
> “Specifically, the LSTM predicts the pen state and produces parameters to form a GMM to estimate the offset distance from previous point.”

---

> > ### Author Response · Authors · 2022-11-16
> > **Author Responses to Reviewer po7R (2/2)**
> >
> > $\textbf{Q6: In section 3.1, the manuscript mentions that corrupted sketches are obtained by “randomly rounding off a certain}$$\textbf{ percentage of strokes”, which would make corrupt sketches always be the same as unfinished sketches. It is not clear}$$\textbf{ however when this corruption technique is used since in section 4.1 another common method is described to be applied for the experiments. }$$\textbf{Was the first method used for training the model and the second used for testing?}$
> >
> > A6: Our approach to masking during training and testing is consistent with that in section 4.1. Our statement in section 3.1 is incorrect and is revised as follows: The corrupted sketch $\bm{X}$ is obtained by masking part of the information in the full sketch $\bm{\tilde{X}}$.
> >
> > $\textbf{Q7: The naming and explanations of the metrics Rec and Ret could be amended to better explain what are the actual metrics }$$\textbf{ used. Is Rec simple top-1 accuracy? Is Ret precision@K? The short names can be used on the tables}$$\textbf{  but please make meaning explicit within the text.}$
> >
> > A7: We have modified the representation of Rec and Ret. The relevant descriptions in section 4.1 are as follows:
> > “$Rec$ indicates whether the recreated sketch $\bm{\hat{S}}$ and its corresponding sketch $\bm{\tilde{S}}$ belong to the same category, e.g. $top-1$ recognition accuracy.”
> > “We use $Ret@k$ to represent $top-k$ retrieval accuracy.”
> >
> > $\textbf{Q8: Writing can be improved specifically on the Introduction. }$
> >
> > A8: We removed “due to the difficulty in learning” from the abstract and introduction in the revision of our paper.
> >
> > We change “it is not always available” in section 1 to “but this information”.
> >
> > We correct “SketchLattice is a lightweight model which limits its performance” in section 1 to “However, during the data processing phase, the lattice approach causes some of the information in the raster sketch image to be lost, thus limiting SketchLattice's performance.”.
> >
> > We modify “Different to […] we expect the network to learn stable sketch representations” in section 1 to “Different from the state-of-the-art graph-structure models, which pass information between nodes to fill in the gaps, we expect that the network to take full advantage of the information in the raster sketch images and learn stable sketch representations in the absence of temporal information.”
> >
> > We change “and even more” in section 1 to “and worse”.
> >
> > We carefully checked the paper and made the following corrections to the typo you pointed out as well as additional typos:
> > Page 2, the second paragraph above the Related Works, "oin" to "in".
> > Page 4, section 3.3, “raseter” to “raster”.
> > Page 5, footnotes, “umbrell” to “umbrella”, “girrafe” to “giraffe”.
> >
> > [3] David Ha and Douglas Eck. A neural representation of sketch drawings. In International Conference on Learning Representations, 2018.

---

### Official Review · Reviewer_hWp5 · 2022-10-24

**Confidence:** 2
**Correctness:** 3
**Technical Novelty And Significance:** 2
**Empirical Novelty And Significance:** 3
**Recommendation:** 5

**Clarity, Quality, Novelty And Reproducibility:**

The paper is well-written and the method shows very good performance. \
The novelty of the method is a bit limited. Regarding reproducibility, I think that the draft is clear enough for reproducing the method but I encourage the authors to release the code, the new Ret metric, and the revised data feeding mentioned in section 4.1.

**Strength And Weaknesses:**

**Strength**

The paper is well-written and the method shows great performance in the experiments.\


**Weaknesses**
- The method's novelty is limited.
- I'm not familiar with the literature but I think the paper might be a better fit for a computer vision conference (like the other baselines).
- In section 4.1 there is a discussion about the information leakage in the dataset preparation in sketchhealer.1 and also changing the Ret evaluation from what the other baselines have previously reported on. How much of the performance gap shown in tables 1 and 2 is coming from these factors, especially the updated Ret metric?
- typo: page 2, the second paragraph above the Related Works, "oin" should be "in"

- Question: Why do the authors think that their method is beating the other baselines with a big margin in Ret ( table 2) but getting closer numbers in Rec (table 3) ?

**Summary Of The Paper:**

This paper proposes LMSER-PIX2SEQ for sketch healing. The method is an encoder-decoder similar to VAE and performs very good with respect to the other baselines.

**Summary Of The Review:**

I think this study is a much better fit for a computer vision conference.

---

> ### Author Response · Authors · 2022-11-16
> **Author Responses to Reviewer hWp5**
>
> Thank you for your time and effort.
>
> $\textbf{Q1: The method’s novelty is limited. }$
>
> A1: See the Answer A1 to Reviewer#1’s (oNd8’s) question Q1.
>
> $\textbf{Q2: The paper is more suitable for computer vision conferences.}$
>
> A2: Our work is focused on learning stable sketch representations for sketch healing. This belongs to representation learning for computer vision. The ICLR 2023 home page has this description: “A non-exhaustive list of relevant topics explored at the conference include: …, representation learning for computer vision and natural language processing, …”. In addition, sketch-rnn [3] and RPCL-pix2seq [2], which are highly relevant to our work, were published in ICLR and Neural Networks rather than in computer vision conferences or journals.
>
> $\textbf{Q3: In section 4.1 there is a discussion about the information leakage in the dataset preparation in sketchhealer.1 and}$$\textbf{ also changing the Ret evaluation from what the other baselines have previously reported on. }$$\textbf{How much of the performance gap shown in tables 1 and 2 is coming from these factors, especially the updated Ret metric?}$
>
> A3: Of the baseline models, RPCL-pix2seq and Sketch-Healer 1.0 [4] have been used for retrieval evaluation. In our paper, we use the retrieval metric from RPCL-pix2seq, i.e., the Ret metric. RPCL-pix2seq’s Ret takes the latent codes of the recreated sketches to retrieve the latent codes of the corresponding full sketches, and it is a more challenging setting than SketchHealer-1.0’s retrieval metric which uses the latent codes of the corrupted sketches directly to retrieve the sketches’ latent codes of the same label. In other words, for a corrupted sketch, there are thousands of results that are correct when using SketchHealer-1.0’s retrieval as a metric, while only one result is correct when using RPCL-pix2seq’s Ret. The change to the masking approach in the SketchHealer-1.0 open-source code is for air comparisons. When testing, the corrupted sketch information should be exactly the same for different model inputs. We illustrate the calculation of Ret more clearly with the figure in the Appendix.
>
> $\textbf{Q4: Why do the authors think that their method is beating the other baselines with a big margin in Ret (table 2) }$$\textbf{ but getting closer numbers in Rec (table 3)?}$
>
> A4: See the Answer A2 to Reviewer#2’s (rRmX’s) question Q2.
>
> $\textbf{Q5: Code release.}$
>
> A5: There is code for our method in the supplementary material. If the paper is accepted, we will release our code.
>
> $\textbf{Q6: Typo.}$
>
> A6: We carefully checked the paper and made the following corrections to the typo you pointed out as well as additional typos:
> Page 2, the second paragraph above the Related Works, "oin" to "in".
> Page 4, section 3.3, “raseter” to “raster”.
> Page 5, footnotes, “umbrell” to “umbrella”, “girrafe” to “giraffe”.
>
>
> [2] Sicong Zang, Shikui Tu, and Lei Xu. Controllable stroke-based sketch synthesis from a self-organized latent space. Neural Networks, 137:138–150, 2021.
>
> [3] David Ha and Douglas Eck. A neural representation of sketch drawings. In International Conference on Learning Representations, 2018.
>
> [4] Guoyao Su, Yonggang Qi, Kaiyue Pang, Jie Yang, and Yi Zhe Song. Sketchhealer a graph-to-sequence network for recreating partial human sketches. In British Machine Vision Conference, 2020.

---

### Official Review · Reviewer_rRmX · 2022-10-27

**Confidence:** 3
**Correctness:** 4
**Technical Novelty And Significance:** 2
**Empirical Novelty And Significance:** 1
**Recommendation:** 5

**Clarity, Quality, Novelty And Reproducibility:**

The paper is generally well written.
Would probably need the full code release from the authors to ensure reproducibility.

**Strength And Weaknesses:**

Strength
 - The self supervised manner of training is pretty clever.
 - The learning of y from the mini task of healing rasterized sketch is also interesting
 - The output is a sequence S of sketch, so should be more useful than rasterized output
 - The qualitative result comparison looks great and shows the strength of the proposed method against others.

Weakness
 - The 2 quantitative metrics used aren’t really satisfying. The R_ret comes from feeding the rasterized version of the generated sketch and the full sketch into the proposed encoder, which was trained for this method to match the two. It doesn’t seem fair to other methods when comparing them. The R_rec is less problematic but the improvement of the proposed method over other methods is very small.

**Summary Of The Paper:**

The paper is an attempt to solve the problem of ‘sketch healing’; trying to recover the original sketch from the corrupted one. The authors propose Lmser-pix2seq, which consist of Lmser block (an encoder-decoder for healing rasterized sketch) and an LSTM decoder (to convert from latent embedding from lmser to full sequence). The evaluation is done on 1) if the recover sketch can be classified as the same class as the complete sketch and 2) if the embedding of the recovered sketch can be used to retrieve the full sketch. The results are better than previous works on both metric, however only slightly in the recognition test.

**Summary Of The Review:**

While I like the cleverness of the approach, I’m a bit concern about the actual problem and its usefulness. Sure sketch healing is great for recover the missing details of the sketch, but in the current setting this is only applying to this specific corruption method. I think it would be great to have another section about generalization for example how this can be applied to actual sketch corruption various source, or how much will this help if we use it as data augmentation method for sketch related task. R_rec sort of try to cover this by showing how it can help with the sketch classification task, but the proposed method only gain small amount of improvement. So as it is currently I’m a bit hesitant to recommend acceptance.

Others:
Also, all these discussion about sketch corrupting, sketch healing, etc. seems like it would be a natural extension to apply diffusion model to it.

---

> ### Author Response · Authors · 2022-11-16
> **Author Responses to Reviewer rRmX**
>
> Thank you for your feedback and suggestions.
>
> $\textbf{Q1: The Ret comes from feeding the rasterized version of the generated sketch and the full }$ $\textbf{sketch into the proposed encoder, which was trained for this method to match the two.  }$ $\textbf{It doesn’t seem fair to other methods when comparing them.}$
>
> A1: Our comparison is fair. Ret is computed from each model’s own encoder. For example, a sketch generated by sketch-pix2seq and the corresponding full sketch are fed into the sketch-pix2seq’s CNN encoder but not our Lmser block. For clarity, we have amended “Specifically, for different models, full sketch $\bm{\tilde{S}}$ and recreated sketch $\bm{\hat{S}}$ are rasterized and fed to their own encoder to obtain vectors $\bm{\tilde{\mu}}$ and $\bm{\hat{\mu}}$. We use $\bm{\hat{\mu}}$ to retrieve its corresponding vector $\bm{\tilde{\mu}}$, and $Ret$ is the success rate of retrieval.”
> Also, we illustrate the calculation of Ret more clearly with a figure in the Appendix.
>
> $\textbf{Q2: The improvement in Rec is small.}$
>
> A2: This is because both the state-of-the-art methods and ours are effective in capturing features at the sketch category-level. However, extracting information at the sketch instance-level is more difficult. We take the umbrella in the fifth row from the bottom of Figure 2 as an example. All RPCL-pix2seq’s generated results are easily recognizable, which leads to high Rec. But the generated umbrellas’ shape or the handles’ direction are changed obviously, which always leads to low Ret. The purpose of our work is not only to recreate sketches of the same category (high rec) but also to better preserve the information of the individual sketches (high ret).
>
> $\textbf{Q3: The actual problem and usefulness.}$
>
> A3: We use other masking methods to verify the generalization of our model. We give the healing results using irregular holes for masking in the Appendix. The significance of sketch healing is that the RNN decoder outputs a vector format that helps generate clear, scalable sketches, which is difficult to do with CNN decoders.
>
> $\textbf{Q4: Code release.}$
>
> A4: We will release the code if our paper is accepted.

---

### Official Review · Reviewer_oNd8 · 2022-10-28

**Confidence:** 3
**Correctness:** 3
**Technical Novelty And Significance:** 1
**Empirical Novelty And Significance:** 1
**Recommendation:** 3

**Clarity, Quality, Novelty And Reproducibility:**

This paper is well-written. It's clear and easy to follow, but the novelty may be limited.

**Strength And Weaknesses:**

**Strengths**
(1) The proposed model has reported state-of-the-art performance on existing datasets for the sketch healing tasks.

**Weaknesses**
(1) The proposed model combines LMSER and Pix2seq models for the sketch healing task, which is incremental. Furthermore, the difference from sketch-pix2seq and RPCL-pix2seq is not clear. I understand that the latter models have been proposed for sketch generation, but joining them with a sketch encoder is not really difficult.

**Summary Of The Paper:**

This paper addresses the sketch healing problem. In other words, it addresses the problem of removing corruptions from sketches. In this work, the authors have proposed the least mean square error reconstruction (LMSER) block for the same, which essentially falls within the encoder-decoder paradigm. The model takes a corrupted sketch as input, then the LMSER encoder computes the embeddings of structural patterns of the input, while the decoder reconstructs the original sketch from the latent embeddings. The decoder is constructed by a recurrent neural network which recreates the sketches from the features captured by the LMSER block.


**Summary Of The Review:**

The paper combines two existing models for sketch healing task, whose novelty is incremental.

---

> ### Author Response · Authors · 2022-11-16
> **Author Responses to Reviewer oNd8**
>
> Thanks for your time and feedback.
>
> $\textbf{Q1: The novelty is limited.}$
>
> A1: Our work is not simply about combining Lmser block and pix2seq, but for accurately capturing instance-level features of sketches through sufficient feature fusion. Previous studies have been effective in capturing features at the sketch category level (high Rec) but still not satisfactory at the instance level (low Ret). The sketch instances of the same category may still be very different from each other. The proposed Lmser-pix2seq addresses this issue. We do this by building bi-directional skip connections and forming loops between the encoder and decoder. Such dual nature enables the representation of learning aware of instance-specific patterns. This practice is rare in the sketching community. Compared with other methods, our model significantly improves the Ret metric and slightly improves the Rec metric.
>
> $\textbf{Q2: The differences between sketch-pix2seq, RPCL-pix2seq and Lmser-pix2seq.}$
>
> A2: These three models differ in the problems they address and in the design of the network architecture.
> The sketch-pix2seq [1] is concerned with category-level generation, which leads to a high Rec score. The main function of RPCL-pix2seq [2] is to generate sketches according to given categories or styles. Lmser-pix2seq focuses on capturing features at the sketch instance-level, with the strength of high Ret score.
> For model design, sketch-pix2seq uses a CNN encoder to extract features and an RNN decoder to output recreated sketches. RPCL-pix2seq introduces a CNN decoder on top of sketch-pix2seq and assumes the latent space follows a Gaussian mixture model (GMM). The CNN decoder in RPCL-pix2seq plays a role of regularization and helps the formation of feature concepts. The network architecture of RPCL-pix2seq is similar to the CAE-pix2seq model in section 4.3. Lmser-pix2seq constructs bi-directional skip connections between the CNN encoder and CNN decoder. This allows both the CNN encoder and the CNN decoder to be involved in feature extraction and image reconstruction. Loops obtained based on bi-directional skip connections are also discussed. Experimental results show that two and more cycles are effective in capturing sketch features.
> We have added relevant content to the revision of the paper.
>
> [1] Yajing Chen, Shikui Tu, Yuqi Yi, and Lei Xu. Sketch-pix2seq: a model to generate sketches of multiple categories. arXiv preprint arXiv:1709.04121, 2017.
>
> [2] Sicong Zang, Shikui Tu, and Lei Xu. Controllable stroke-based sketch synthesis from a self-organized latent space. Neural Networks, 137:138–150, 2021.

---

### Decision · Program_Chairs · 2023-01-20

**Decision:**

Reject

**Justification For Why Not Higher Score:**

The novelty is incremental, and evaluation metrics need to be updated to be more fair.

**Justification For Why Not Lower Score:**

n/a

**Metareview: Summary, Strengths And Weaknesses:**

**Summary**: The paper is an attempt to solve the problem of ‘sketch healing’; trying to recover the original sketch from the corrupted one. The authors propose Lmser-pix2seq, which consist of Lmser block (an encoder-decoder for healing rasterized sketch) and an LSTM decoder (to convert from latent embedding from lmser to full sequence). The evaluation is done on 1) if the recover sketch can be classified as the same class as the complete sketch and 2) if the embedding of the recovered sketch can be used to retrieve the full sketch.

**Strengths**: The paper is well-written and the method shows great performance in the experiments.

**Weaknesses**: The following weaknesses were identified by the reviewers:

1. incremental novelty - combining LMSER and Pix2seq. [oNd8, hWp5]
2. the difference between sketch-pix2seq and RPCL-pix2seq is unclear. [oNd8]
3. The R_ret metric is not fair -- the generated sketch and full sketch are compared using the model encoder. [rRmX]
4. The results using R_rec metric show only small improvements. [rRmX]
5. Only applied to a specific corruption method. How does the method generalize to actual sketch corruption from various sources? [rRmX]
6. how much of the performance improvement is related to the change in the Ret metric and the information leakage? [hWp5]
7. Why are the improvements much better for Ret metric (table 2) than the Rec metric (table 3)? [hWp5]
8. comparison with SketchLattice is flawed (input requirement is not the same). [po7R]
9. Other models presented results aligned with the original SketchLattice paper. [po7R]
10. How well does it work when only trained with CNNs (without vector decoder)? [po7R]
11. various improvements in presentations [po7R]
12. unclear if the "unfinished sketch" corruption is actually used. [po7R]

**Discussion:** During the discussion, one reviewer remained positive about the paper, while others did not change their ratings. The major issues were 1) the incremental novelty, and 2) the suitability of the proposed retrieval metric. Regarding novelty, the paper mainly combines LMSER and pix2seq. Regarding the retrieval metric, the AC pointed out that the Ret metric uses each model's own encoder to do the evaluation, which means that each compared method is using a different retrieval model. This doesn't seem fair because some models may have better encoders (more complex architectures), or be specifically trained to have a retrievable internal representation. To be fair, the same retrieval model (distinct from all the sketch reconstruction methods) should be used for all methods to evaluate Ret. It was also pointed out that using a CNN (as asked by Reviewer po7R) actually outperformed the proposed method for small mask percentages on retrieval, and outperforms all methods at Rec. Reviewer po7R pointed out that using CNN might not be suitable for sketch healing, since the results will be blurry. Probably more suitable evaluation metrics are needed.


**Summary Of Ac-Reviewer Meeting:**

n/a